# ELF5 modulates the estrogen receptor cistrome in breast cancer

Catherine L. Piggin[1,2¤a], Daniel L. Roden[1,2], Andrew M. K. Law[1,2], Mark P. Molloy[3¤b], Christoph Krisp[3¤c], Alexander Swarbrick[1,2], Matthew J. Naylor[1,2,4], Maria Kalyuga[1,2], Warren Kaplan[1,2], Samantha R. Oakes[1,2], David Gallego-Ortega[1,2], Susan J. Clark[1,2], Jason S. Carroll[5], Nenad Bartonicek[1,2], Christopher J. Ormandy[1,2]*

**1** Garvan Institute of Medical Research and The Kinghorn Cancer Centre, Victoria Street Darlinghurst Sydney, NSW, Australia, **2** St Vincent's Clinical School, Faculty of Medicine, UNSW Sydney, Australia, **3** Australian Proteome Analysis Facility, Macquarie University, Sydney, Australia, **4** School of Medical Sciences, Faculty of Medicine and Health, The University of Sydney, Sydney, Australia, **5** Cancer Research UK Cambridge Research Institute, Li Ka Shing Centre Robinson Way, Cambridge, United Kingdom

¤a  Current address: Department of Biomedicine (DBM), University Hospital, Basel, Switzerland
¤b  Current address: Northern Clinical School Royal North Shore Hospital, University of Sydney, Sydney, Australia
¤c  Current address: University Medical Center Hamburg-Eppendorf, Institute of Clinical Chemistry and Laboratory Medicine, Mass Spectrometric Proteome Analysis, Hamburg, Germany
*  c.ormandy@garvan.org.au

**Data Availability Statement:** Raw data is available via ArrayExpress E-MTAB-7641and Proteome exchange PXD013349.

**Funding:** This work was supported by grants from the NHMRC Australia (GNT1068753, GNT1043400,

## Abstract

Acquired resistance to endocrine therapy is responsible for half of the therapeutic failures in the treatment of breast cancer. Recent findings have implicated increased expression of the ETS transcription factor *ELF5* as a potential modulator of estrogen action and driver of endocrine resistance, and here we provide the first insight into the mechanisms by which ELF5 modulates estrogen sensitivity. Using chromatin immunoprecipitation sequencing we found that ELF5 binding overlapped with FOXA1 and ER at super enhancers, enhancers and promoters, and when elevated, caused FOXA1 and ER to bind to new regions of the genome, in a pattern that replicated the alterations to the ER/FOXA1 cistrome caused by the acquisition of resistance to endocrine therapy. RNA sequencing demonstrated that these changes altered estrogen-driven patterns of gene expression, the expression of ER transcription-complex members, and 6 genes known to be involved in driving the acquisition of endocrine resistance. Using rapid immunoprecipitation mass spectrometry of endogenous proteins, and proximity ligation assays, we found that ELF5 interacted physically with members of the ER transcription complex, such as DNA-PKcs. We found 2 cases of endocrine-resistant brain metastases where ELF5 levels were greatly increased and ELF5 patterns of gene expression were enriched, compared to the matched primary tumour. Thus ELF5 alters ER-driven gene expression by modulating the ER/FOXA1 cistrome, by interacting with it, and by modulating the expression of members of the ER transcriptional complex, providing multiple mechanisms by which ELF5 can drive endocrine resistance.

GNT1063559), Cancer Council NSW (PG11-07, RG17-02, RG19-07), Cancer Institute NSW (DG00625), RT Hall Trust, Mostyn Family Foundation, Cue Clothing Co., Estee Lauder Australia and Sutton's Motors Sydney. These sponsors played no role in study design, data collection and analysis, decision to publish or preparation of the manuscript.

**Competing interests:** The authors have declared that no competing interests exist.

## Author summary

Two thirds of breast cancers are initially treated with endocrine therapy because they are likely to rely on estrogen for their proliferation. Understanding why this therapy ultimately fails in 2/3 of cases offers the chance of durable treatment. In 2012 we hypothesised that normal developmental cell fate decisions taken by mammary progenitor cells persist in tumours that are maintained by instances of a cancerous progenitor, and that a change in the relative influence of the two major transcription factors that drive progenitor cell fate, ER to specify the hormone sensing lineage, and ELF5 to specify the ER- alveolar lineage, may allow a cancer to shift control of proliferation from estrogen to ELF5. Here we show that these transcription factors are often co located at super enhancers, enhancers and at the promoters of differentially regulated genes. When the levels of ELF5 were increased this caused ER and its pioneer factor FOXA1 to move to new regions of the genome associated with resistance to hormonal therapy. We also showed that ELF both regulated and bound directly to members of the ER-transcriptional complex. These findings provide the first indication of the mechanisms by which an increase in ELF5 could drive the acquisition of endocrine resistance.

## Introduction

Estrogen receptor (ER) positive breast cancer is initially treated using endocrine therapy to withdraw estrogen, destroy its receptor, or alter ER-driven transcription [1]. ER+ breast cancer exhibits a unique and significant long-term risk of distant recurrence, characterized by renewed metastatic activity and resistance to endocrine therapy [2]. Resistance occurs via multiple mechanisms [3, 4]. Mutations causing constitutive ER activity or hypermethylation of ER enhancers occur in response to estrogen withdrawal [5, 6]. Activation of signalling events downstream from ER can enhance or replace ER activation [7]. Repositioning of the ER transcriptional complex can occur [8–13], altering both the expression of individual genes and transcriptional programs [9]. These events ultimately regulate the basic cell-cycle and cell-death machinery, and disruptions here can also cause endocrine resistance [4]. Interventions at all of these points provide opportunities to treat endocrine-resistant disease.

ELF5 is an ETS transcription factor that drives a cell fate decision by mammary progenitor cells, causing them to establish the ER negative (ER-) cell lineage responsible for alveolar development and milk production [14]. The hormones of pregnancy act on the ER+ hormone sensing cells to cause secretion of RANKL and other paracrine regulators, which cause the expansion of the stem cell compartment [15], and as they differentiate to produce progenitor cells, epigenetic mechanisms cause *ELF5* to be expressed [16], which is induced further by RANKL, so driving further progenitor cell differentiation to make the mature milk-producing cells of the alveoli [17, 18]. Forcing *ELF5* expression causes ER+ breast cancer cells to adopt gene expression patterns more like those seen in the ER- subtypes [19]. ELF5 levels increase when MCF-7 breast cancer cells are made resistant to endocrine therapies, and these cells also become dependent on ELF5 for their proliferation [19]. In mice, forcing ELF5 expression results in mammary tumor angiogenesis and an increase in lung metastasis via regulation of the innate immune system [20]. Thus ELF5 is associated with the two key aspects of progression to the lethal phenotype in ER+ breast cancer, acquisition of endocrine resistance and renewed metastatic activity. This conclusion is supported by clinical data. In luminal A breast cancer patients treated with tamoxifen, high ELF5 expression strongly predicted early treatment failure and disease progression [20].

In 2012 we proposed that luminal breast cancer cells may retain some of the cell-fate plasticity of the progenitor cell population seen in normal development, and that the expressed cancer phenotype results, at least in part, from the balance between ELF5 and ER influence [19]. Cancer cells may become insensitive to estrogen when ELF5 increases. Conditions that upregulate ELF5 expression, for example continuous endocrine therapy [19], or increased exposure to RANKL once resident in bone [17], may drive phenotypic conversion to an endocrine-resistant state via the remaking of the cell-fate decisions taken by the tumour initiating cells. To do this ELF5 must modulate ER-driven gene expression, but the mechanism is unknown. Here we report the result of a search for potential interactions between ELF5 and ER, using ChIP-seq, RIME and RNA-seq, in MCF-7 breast cancer cells, the model which alone has provided most of our understanding of the mechanisms of endocrine resistance [21]. We show that induction of ELF5 alters the ER cistrome and that this alteration is similar to that caused by the acquisition of endocrine resistance. We also show that ELF regulates, and interacts with, members of the ER transcriptional complex. These findings provide the first steps toward elucidation of the molecular mechanisms by which ELF5 modulates ER driven gene expression, to promote endocrine resistance.

## Results

### ELF5 binding to the genome overlaps with FOXA1 and ER binding

We used doxycycline-inducible expression of human ESE2B, the ELF5 isoform expressed in mammary gland [22], combined with ChIP-seq from MCF-7 and T47-D breast cancer cells, to search for a transcriptional mechanism that allows ELF5 to modulate estrogen-driven gene expression. A series of quality control measures showed that our expression system [19] did not induce ELF5 levels beyond physiological levels (S1A Fig) and at the time points used ER and FOXA1 levels remained unchanged by induction of ELF5 (S8B and S8C Fig). MCF-7 ChIP-seq data set (ELF5, FOXA1, ER and H3K4Me3) had sufficient read depth with a low proportion of errors (S1B Fig), high quality scores (S1C Fig), low duplication levels (S1D Fig), and consistent GC content (S1E Fig). Ten percent of reads were called in peaks by MACS (S1F Fig) and signal strength was comparable across the four replicates, and the four ChIPs (S1G Fig). A principal components analysis showed that requiring binding peaks called by MACS [23] in three of the four replicates provided high discriminatory power without overly limiting peak numbers (S1H Fig) and a consensus of a peak in 3 of 4 replicates was used to define genomic binding sites. We found 28363 ELF5 peaks, many in the distal intergenic regions but about half in promoter regions (Fig 1A), with reduced frequency as distance from the transcriptional start site (TSS) increased (Fig 1B). An analysis using the MEME Suite [24], of DNA sequences at ELF5 binding peaks found that a consensus ELF5 binding motif was present with high probability, as were consensus binding motifs for CTCF, FOXA1 and ER (Fig 1C). GREAT [25] was used to identify the function of genes potentially regulated by ELF5. Using the Molecular Signature Data Base (MSigDB) gene sets, perturbations such as response to treatment with estrogen, tamoxifen, EGF and chemotherapeutic drugs were significantly enriched (Fig 1D and S2A Fig), as were processes including peptidyl-proline modification and mammary morphogenesis, and pathways such as signal transduction and extracellular matrix, the interferon response and nuclear receptor action. The ELF5 peaks in MCF-7 cells overlapped very substantially (80%) with ELF5 ChIP-seq peaks in T47-D cells (S2B Fig), showing a similar effect of ELF5 in a different cell line.

Meme Suite also identified three sequences with homology to Alu transposable elements that overlapped ELF5 binding peaks and recurred in a variety of combinations (S3A Fig). Further investigation using RepeatMasker showed that mammalian inter-dispersed repeats (MIR)

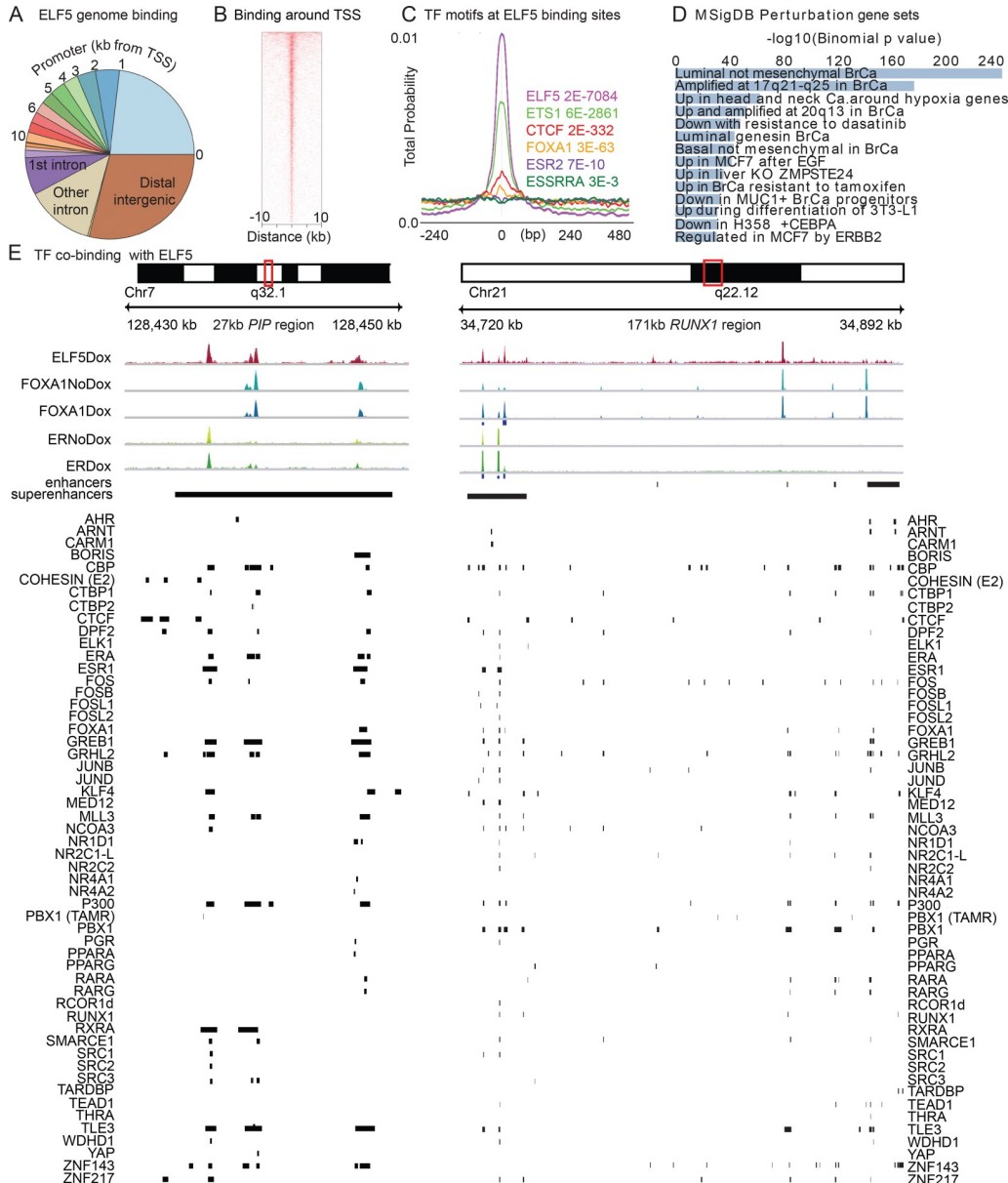

**Fig 1. Genomic binding of ELF5 is co-located with ER and FOXA1, and occurs at promoter, enhancer and super enhancer regions.** Panel **A**, proportion of ELF5 binding to promoter regions at the indicated distance from the transcription start site (TSS) and to non-promoter regions. Panel **B**, distribution of ELF5 binding relative to the TSS. Panel **C**, positional distribution in base pairs (bp) (CentriMO plot) of the indicated transcription factor (TF) binding motifs relative to all ELF5 ChIP-binding sites centred at 0. Total probability for all curves in the figure is 1. Panel **D** Functional annotation by GREAT of ELF5 binding sites. Panel **E**, Transcription factor binding at enhancers and super enhancers near *PIP* and *RUNX1* genes comparing ELF5, FOXA1 and ER binding with and without induction of ELF5 using doxycycline (DOX), in relation to the binding of the indicated transcription factors reported by ChIP-seq experiments in MCF-7 cells. Variation among replicates indicated by color shading. Blue boxes under FOXA1Dox and ERDox tracks indicate peaks called as statistically significantly increased (FDR<0.05) by DiffBind.

of the SINE class were most the frequently found repetitive element at ELF5 peaks (S3B Fig), and that a consensus ETS motif was present within these repeats (S3C Fig). SINE elements were enriched close to Elf5 peaks but not at a distance (S3D Fig). Using Repbase to quantify this finding [26], we found that ELF5 peaks frequently overlapped a repetitive element of

different subtypes (S3E Fig). In transcriptionally active sites, such as highly occupied target binding regions (HOT) [27], enhancers and super enhancers, the odds ratio diverged from 1, and in genes that showed differential expression in response to ELF5 induction, there was significant enrichment of mammalian interspersed repeats (MIRs) (Odds ratio = 2, p = 8E-281) (S3E Fig). Thus ELF5 binds to the genome at ETS sites, many of which are located in repetitive elements [28]. These elements are rich in transcription factor binding sites and are proposed to act as transposable enhancers that drive genome evolution [29], by placing the binding of key transcription factors such as MYC, SP1 p53, RAR, ER, and now ELF5, into new genomic contexts [30]. Thus ELF5 is an additional transcriptional influence driving genome evolution by MIR elements and these sites remain active in the regulation of gene expression.

We compiled public data from 134 MCF-7 ChIP-seq experiments (SDoc2 Text) and used correlation coefficients to identify sets of transcription factors that co-bind with ELF5, within genomic sub-regions such as enhancers, super enhancers [31], or genes with differential expression. The sets were visualized using UpSet [32] for the most frequent examples (S4A–S4C Fig), showing the number of times co-binding of the indicated set of transcription factors was observed. ELF5 binds to the genome with different groups of transcription factors at genomic loci with different functions. In the promoters of genes showing differential expression in response to elevated ELF5 the most frequent binding pattern was ELF5 in combination with a few transcription factors. For example, in 119 instances of genomic ELF5 binding to the promoters of differentially expressed genes, ELF5 bound alone with FOXA1 in 7 instances, with FOS alone in 6 instances, and with up to 6 transcription factors in 60 instances, most unique in their combinations. Factors binding with ELF5 included ER-transcription complex members GREB1, TRPS1, SRC2, TLE3, MLL3, JUN, CBP, endocrine receptors RARG, RARA, RXRB, orphan nuclear receptor NR2C1, co-repressor RCOR1, co-activators NCOA2 (SRC2) and p300, and chromatin remodeller SMARCE 1 (S4A Fig). A very different pattern occurred at super enhancers (S4B Fig). MED1 is often used to define super enhancers and MED1 is frequently found to be a member of the 20+ transcription factor set including many members of the ER transcriptional complex such as ER, FOXA1, MLL3, GREB1, SRC2,TLE3 etc. NIPBL and Cohesin are also frequent members of this set, and are involved in enhancer-promoter communication. Fig 1E shows two instances of enhancers or super enhancers located near the *PIP* and *RUNX1* genes. Near *PIP* ELF5 bound to three sites that also bound FOXA1 and a third site that bound ER. All three sites showed overlapping binding of a large number of transcription factors. The super enhancer near *RUNX1* bound ELF5, and Diffbind [12] showed increased levels of FOXA1 and ER binding (blue squares), when ELF5 levels were induced. Enhancers in this region also bound both ELF5 and FOXA1 (S4C Fig). Enhancers showed patterns similar to promoters or super enhancers, suggesting either mis-annotation or dual functionality. Thus ELF5 binds to super enhancers together with a large number of transcription factors, but at the promoters of differentially expressed genes with a much smaller group. Many of the transcription factors with overlapping binding with ELF5 are members of the ER transcriptional complex.

## ELF5 driven changes in gene expression

We used RNA-seq to examine the effects of induction of ELF5 on gene expression. Analysis of our RNA-seq data using Limma-Voom found 256 up-regulated genes and 291 down-regulated genes using the cut off values of FDR <0.05 and FC >1.5. The GO consortia functions of development, morphogenesis, differentiation and cell cycle were most prominently diminished by induction of ELF5, while functions of amino acid biology, metabolic, catabolic, and translation were enhanced (Fig 2A). The set of genes showing differential expression in T-47D

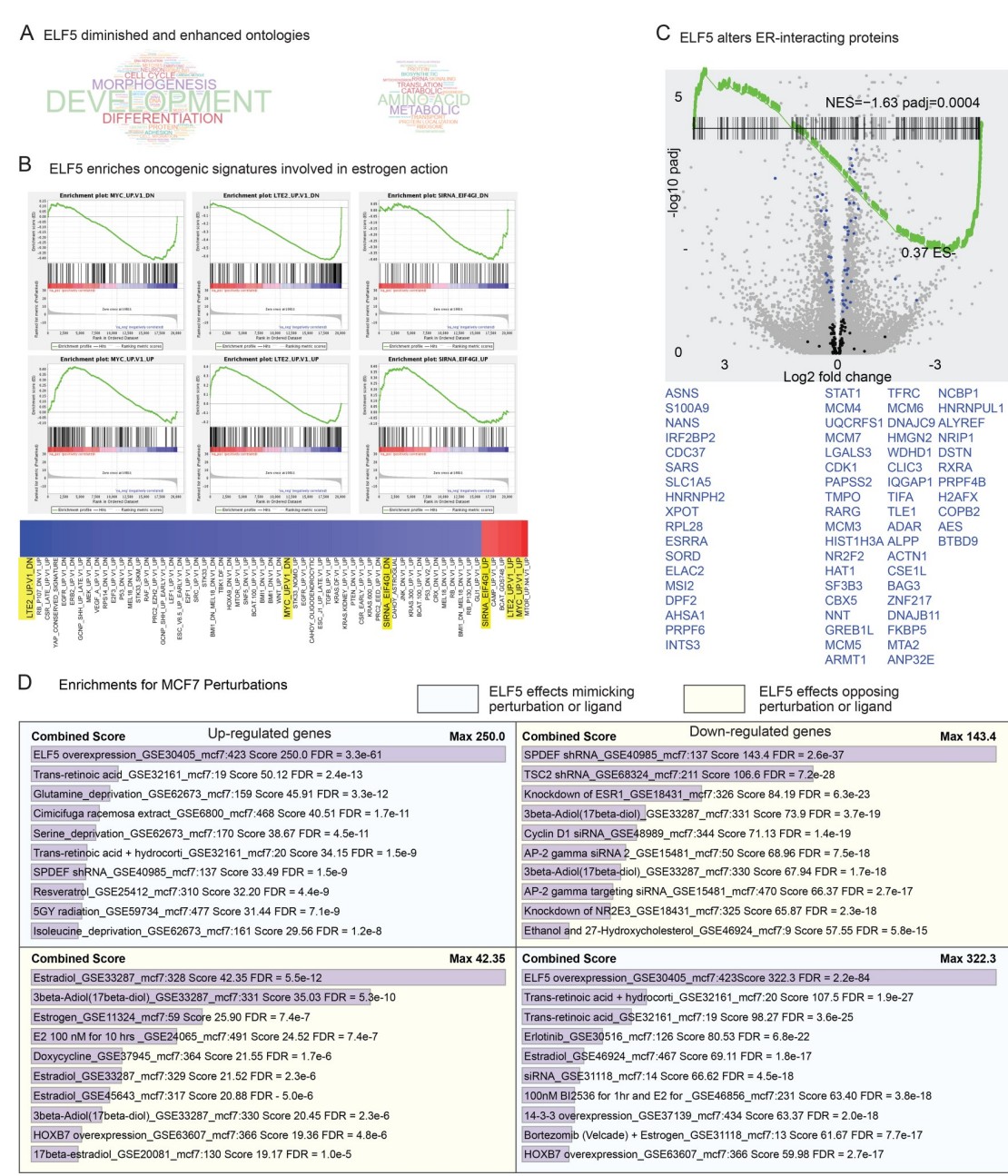

**Fig 2. ELF5 transcriptional effects measured by RNA-seq.** Panel **A**, GO ontologies enriched in the RNA-seq data set by induction of ELF5, font size relative to enrichment score. Panel **B**, GSEA of RNA-seq data using the MSigDB oncogenic signatures gene sets. Heatmap shows all enriched sets arranged by nominal enrichment score, with the sets where pairs showed opposite enrichment highlighted in yellow and shown in detail above. Panel **C**, GSEA of a set of genes formed from proteins that interact with ER, overlaid on a volcano plot of all genes in the RNA-seq analysis. Genes with increased expression (left hand side-trailing edge) or decreased expression (right hand side- leading edge) of the enrichment are highlighted and listed, in blue, in their corresponding order. Panel **D**, enriched gene sets related to the indicated perturbations in MCF7 cells. MCF7 up or down indicates that the perturbation up- or down-regulates genes in the set. The top 10 MCF7 gene set overlaps with significantly up-regulated genes (left) and down-regulated genes (right) from the MCF7-ELF5 RNA-seq experiment are shown.

breast cancer cells in response to induction of ELF5 [19] was very significantly (padj = 1.6E-5 GSEA) enriched, demonstrating that the effects seen in MCF-7 cells also occurred in another cell line.

Using GSEA with the MSigDB oncogenic signature sets we observed 3 conditions that produced changes in gene expression similar to induction of ELF5 (Fig 2B). Genes that went up or down in response to long term estrogen deprivation also went up or down in response to ELF5, consistent with ELF5 promoting an estrogen insensitive phenotype. Similarly, induction of *ELF5* replicated the effects of induction of *MYC. MYC* is a direct transcriptional target of ELF5 and showed a fold change of 1.43 with FDR of 2E-4. ELF5 also produced effects seen in a model of EIF4GI function, a translation initiating factor linking nutrient starvation to inhibition of MTOR and cell proliferation [33]. Using GSEA we examined the expression of a set of genes producing proteins identified as interacting with ER [34]. From this set 18 genes had increased expression (named on the left-hand side Fig 2C) and 49 showed decreased expression (named on the right-hand side Fig 2C). Thus induction of ELF5 caused changes in the expression of genes encoding many members of the ER transcriptional complex. The Enrichr tool [35], showed that perturbations that opposed the changes in gene expression caused by the induction of ELF5 were dominated by treatment with various estrogens (Fig 2D). Using the MSigDB Hallmark sets, functions such as late estrogen response, interferon action, epithelial to mesenchymal transition and aspects of proliferation were significantly depleted, while early estrogen responses were increased together with many aspects of metabolism (S6A Fig), and similar effects on sets of estrogen regulated genes were found in the MSigDB C2 _all gene sets (S6B Fig). The GSEA was visualized using the Enrichment Map plugin for Cytoscape [36, 37] (S5 Fig, a scalable PDF that can be zoomed once downloaded), which presents a comprehensive picture of these transcriptional responses to induction of ELF5. Overall the RNA-seq analysis shows that induction of ELF5 opposed estrogen-driven patterns of gene expression, consistent with the promotion of an estrogen insensitive phenotype.

## ELF5 causes FOXA1 and ER to bind to new regions of the genome in the same pattern caused by the acquisition of endocrine resistance

The Enrichr tool [35] was used to identify transcription factor motifs in the promoters of genes showing altered gene expression in response to ELF5. These promoters were enriched for ER and FOXA1 binding motifs as well as ETS motifs (S6C–S6F Fig). We examined the co-occurrence of binding peaks in our ChIP data called present by MACS for ELF5, FOXA1 and ER using Venn analysis (S7A Fig with UpSet analysis in S7B Fig). There was a large overlap in the binding of these transcription factors, with 1244 genomic sites showing overlapping-binding of ELF5, FOXA1 and ER, 7703 with overlapping-binding of ELF5 and FOXA1, and 194 with overlapping-binding of ER and ELF5. Induction of ELF5 caused FOXA1 to bind to 5875 new genomic sites and ER to bind to 1616 new sites. The probability that an ELF5 motif was present at sites of new binding was increased, but not where binding was lost (S7C and S7D Fig). There was no difference in the increase or decrease of FOXA1 binding sites at enhancers compared to super enhancers, and whether they were proximal or distal to the regions they regulate (S7E Fig), nor was there any variation in the sequence of the ELF5 binding motif at different genomic regions (S7F Fig). We used DiffBind to statistically test these associations of ELF5 with FOXA1 and ER. MACS simply calls peaks present or absent, but DiffBind determines if a peak has shown a statistically significant increase or decrease in binding. At a false discovery rate of <0.05, induction of ELF5 caused ER and FOXA1 binding to increase at 404 and 503 sites (Fig 3A, top panel, UpSet analysis and corresponding Venn), and to decrease at 5 and 293 sites respectively (Fig 3A lower panel). The patterns of FOXA1 and ER genome binding prior to, and following, induction of ELF5 are shown in Fig 3B, with enriched sites indicated in green. GREAT showed that the predominant function of these altered sites was estrogen action and endocrine resistance (Fig 3C). Using genome DNAase1 sensitivity data

**A** ELF5 binding overlaps with FOXA1 and ER and ELF5 alters FOXA1 and ER binding.

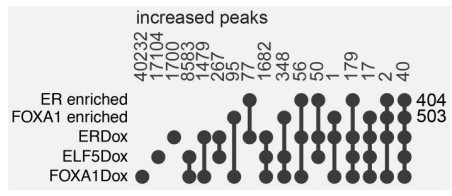

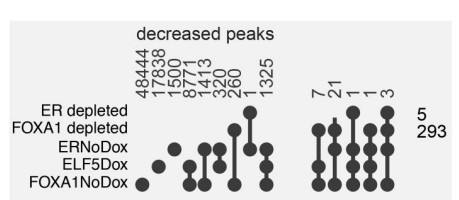

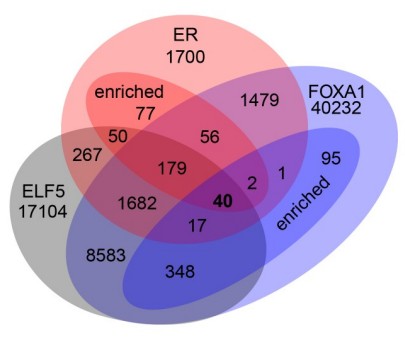

**B** ELF5 increased FOXA1 and ER binding

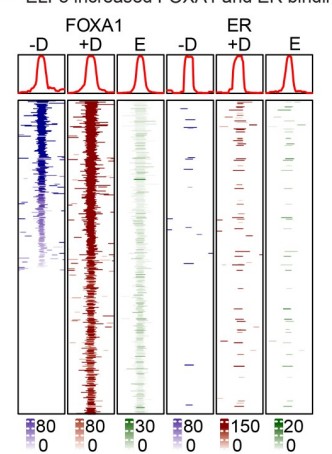

**C** ELF5 caused increased FOXA1 and ER binding at regions associated with estrogen action and hormone resistance.

FOXA1 increased

| Gene Set Name | Genes | Overlap | FDR |
|---|---|---|---|
| GOZGIT_ESR1_TARGETS_DN | 781 | 38 | 1.90E-15 |
| GRAESSMANN_APOPTOSIS_BY_DOXORUBICIN_DN | 1781 | 54 | 3.39E-14 |
| PILON_KLF1_TARGETS_DN | 1972 | 56 | 9.34E-14 |
| DODD_NASOPHARYNGEAL_CARCINOMA_UP | 1821 | 53 | 1.95E-13 |
| CREIGHTON_ENDOCRINE_THERAPY_RESISTANCE_3 | 720 | 33 | 5.32E-13 |
| DACOSTA_UV_RESPONSE_VIA_ERCC3_DN | 855 | 35 | 1.65E-12 |
| NUYTTEN_EZH2_TARGETS_UP | 1037 | 38 | 2.80E-12 |
| MASSARWEH_TAMOXIFEN_RESISTANCE_UP | 578 | 28 | 1.56E-11 |
| DUTERTRE_ESTRADIOL_RESPONSE_24HR_DN | 505 | 26 | 2.98E-11 |
| CREIGHTON_ENDOCRINE_THERAPY_RESISTANCE_5 | 482 | 25 | 7.10E-11 |
| BLALOCK_ALZHEIMERS_DISEASE_UP | 1691 | 46 | 9.85E-11 |
| GRYDER_PAX3FOXO1_ENHANCERS_IN_TADS | 975 | 34 | 2.03E-10 |
| NUYTTEN_NIPP1_TARGETS_DN | 848 | 31 | 6.40E-10 |
| CHICAS_RB1_TARGETS_CONFLUENT | 567 | 25 | 1.79E-09 |
| LEE_BMP2_TARGETS_UP | 745 | 28 | 3.68E-09 |

ER increased

| Gene Set Name | Genes | Overlap | FDR |
|---|---|---|---|
| GOZGIT_ESR1_TARGETS_DN | 781 | 39 | 5.85E-23 |
| MEISSNER_BRAIN_H3K4ME3_AND_H3K27ME3 | 1069 | 35 | 1.03E-14 |
| SMID_BREAST_CANCER_BASAL_DN | 701 | 26 | 1.06E-11 |
| CREIGHTON_ENDOCRINE_THERAPY_RESISTANCE_1 | 528 | 23 | 1.11E-11 |
| GRAESSMANN_APOPTOSIS_BY_DOXORUBICIN_DN | 1781 | 39 | 2.32E-11 |
| BHAT_ESR1_TARGETS_NOT_VIA_AKT1_UP | 211 | 16 | 2.32E-11 |
| HALLMARK_ESTROGEN_RESPONSE_EARLY | 200 | 15 | 1.54E-10 |
| DACOSTA_UV_RESPONSE_VIA_ERCC3_DN | 855 | 26 | 3.82E-10 |
| ZWANG_TRANSIENTLY_UP_BY_1ST_EGF_PULSE | 1839 | 37 | 8.83E-10 |
| GRYDER_PAX3FOXO1_ENHANCERS_IN_TADS | 975 | 26 | 5.71E-09 |
| CHARAFE_BREAST_CANCER_LUMINAL_VS_BASAL_UP | 380 | 16 | 9.40E-08 |
| CREIGHTON_ENDOCRINE_THERAPY_RESISTANCE_3 | 720 | 21 | 1.00E-07 |
| MASSARWEH_TAMOXIFEN_RESISTANCE_UP | 578 | 19 | 1.00E-07 |
| DUTERTRE_ESTRADIOL_RESPONSE_6HR_UP | 229 | 13 | 1.05E-07 |
| FARMER_BREAST_CANCER_APOCRINE_VS_BASAL | 330 | 14 | 8.85E-07 |

**D** ELF5 and hormone resistance cause similar changes to FOXA1 and ER binding.

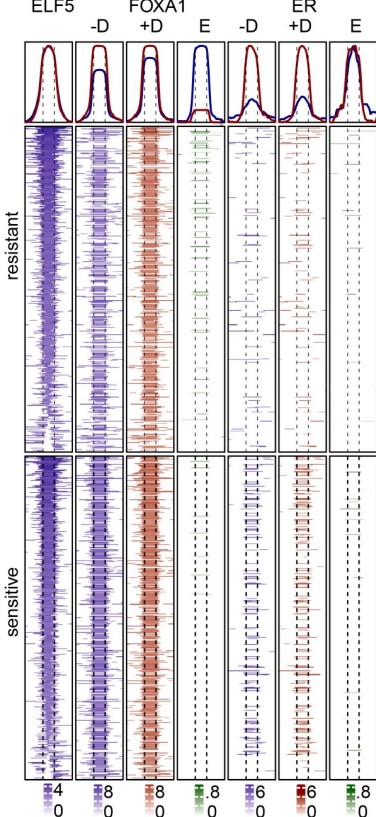

**Fig 3. Induction of ELF5 alters FOXA1 and ER binding to the genome.** Panel **A**, UpSet presentation of DiffBind output that identified statistically significant increases in FOXA1 or ER binding in response to induction of ELF5 with doxycycline (Dox). Corresponding Venn diagram for increased (enriched) peaks is shown. Lower panel, decreased peaks. Panel **B**, FOXA1 peaks that were increased by induction of ELF5 (+D, red) are plotted, centred on the FOXA1 peak, with the corresponding binding of FOXA1 at that location without induction of ELF5 (-D, blue). The resulting differential binding is shown in green. The binding of ER at these FOXA1 sites is shown with (+D, red), and without (-D, blue), with differential binding in green. Histograms show cumulative intensity on the y axis over the corresponding genomic region on the x axis. Color scale denotes log 10 of MACS (-D and +D columns) or DiffBind scores (E columns). Panel **C**, GREAT analysis of binding site function for increased binding sites for FOXA1 and ER. Blue indicates sets involved in estrogen action. Panel **D**, FOXA1 and ER binding sites that are specific in MCF-7 cells that were either sensitive to, or made resistant to, endocrine therapy, with corresponding ELF5 binding intensity. Histograms compare overall intensity with (+D) and without (-D) induction of ELF5 and green shows the enrichment (E) in response to ELF5. Color scale denotes log 10 of MACS (-D and +D columns) or DiffBind scores (E columns).

[38] that defined regions of open chromatin in MCF-7 cells, we found that of the 503 FOXA1 sites enriched by induction of ELF5, only 139 (28%) occurred at regions of open chromatin, indicating that most of the enriched FOXA1 sites occurred at previously closed chromatin. Thus induction of ELF5 causes FOXA1 to bind to closed chromatin, consistent with its established pioneering activity.

The increased FOXA1 binding sites caused by induction of ELF5 overlapped significantly with FOXA1 binding sites increased by the acquisition of endocrine resistance [39] (Chi-squared 8E-16) compared to endocrine sensitive MCF-7 (Fig 3D). Comparison to ER sites showed a similar trend (p = 0.1). There was only 1 decreased FOXA1 site that overlapped with the decreased FOXA1 sites in resistant MCF-7 cells, and there were only 7 ER sites in total that were decreased by induction of ELF5, making it unlikely that the ELF5-induced decrease in FOXA1 or ER binding is involved in endocrine resistance. These results demonstrate that induction of ELF5 increases FOXA1 binding to the same sites that the acquisition of endocrine resistance does, providing a genomic mechanism by which increased ELF5 can drive the acquisition of endocrine resistance.

## ELF5 drives changes in gene expression via the alteration of FOXA1 and ER binding to the genome

To determine whether genes that bound ELF5, FOXA1 and ER in various combinations showed differential gene expression in response to induction of ELF5, we combined our RNA-seq and ChIP-seq data, by plotting the positions of each gene in a gene set formed based on the ChIP data, in their order based on their differential expression in the RNA-seq data, and then calculated an overall enrichment score (NES) and p value for clustering (enrichment) at either end of the differential expression spectrum. This technique does not require the use of arbitrary cut offs to define changed level of expression. These plots were characterised by genes showing both increased and decreased gene expression, indicated by accumulation of genes at either end of the enrichment plot. For gene sets formed based on the presence of various combinations of ELF5, ER and FOXA1 binding, we frequently observed enrichments, showing a transcriptional effect of these transcription factors (Fig 4A). We used a similar approach to determine whether the sets of genes that showed increased FOXA1 or ER genomic binding in response to induction of ELF5, also showed altered gene expression. Where FOXA1 binding increased we observed increased gene expression, and where FOXA1 genomic binding was depleted, reduced gene expression was observed (Fig 4B). Thus the change in FOXA1 and ER binding caused by ELF5 resulted in altered gene expression.

Six genes, (*CD36* [40], *CUEDC1* [41], *LAMP3* [42], *SDCBP* [43], *LTBP2* [44], *PIP* [45]), of the set of 26 differentially expressed genes with enriched or depleted binding of FOXA1 or ER in response to induction of ELF5, are implicated endocrine resistance, and nearly all genes in this set appear in one or more gene sets derived from estrogen action, breast cancer subtype or endocrine resistance. Fig 4C shows ChIP-seq and RNA-seq results for one example, *PIP*. Increased FOXA1 binding was called by DiffBind at two sites (pink highlights), the first of which corresponded to a region binding a large suite of transcription factors associated with the ER transcriptional complex, such as GRHL2, MLL3, TEAD etc. Though not called by Diff-Bind, examination of the plots shows increased ER binding also at this site and elsewhere. *PIP* RNA-seq (top panels) showed a clear increase in transcript reads.

## ELF5 binds directly to members of the ER transcriptional complex

We conducted an unbiased search for ELF5 binding partners using rapid immunoprecipitation mass spectrometry (RIME), [34, 46] identifying 74 interactions (S1 Table). ELF5 was

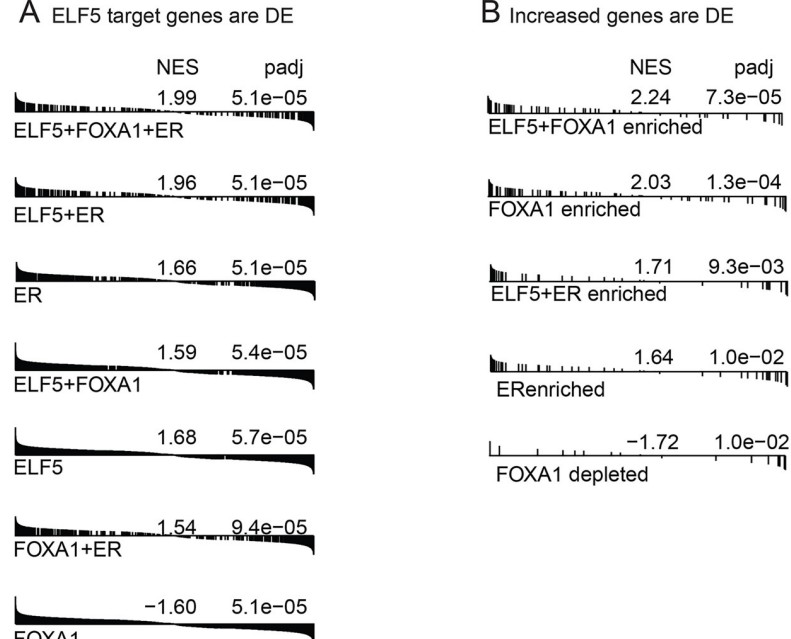

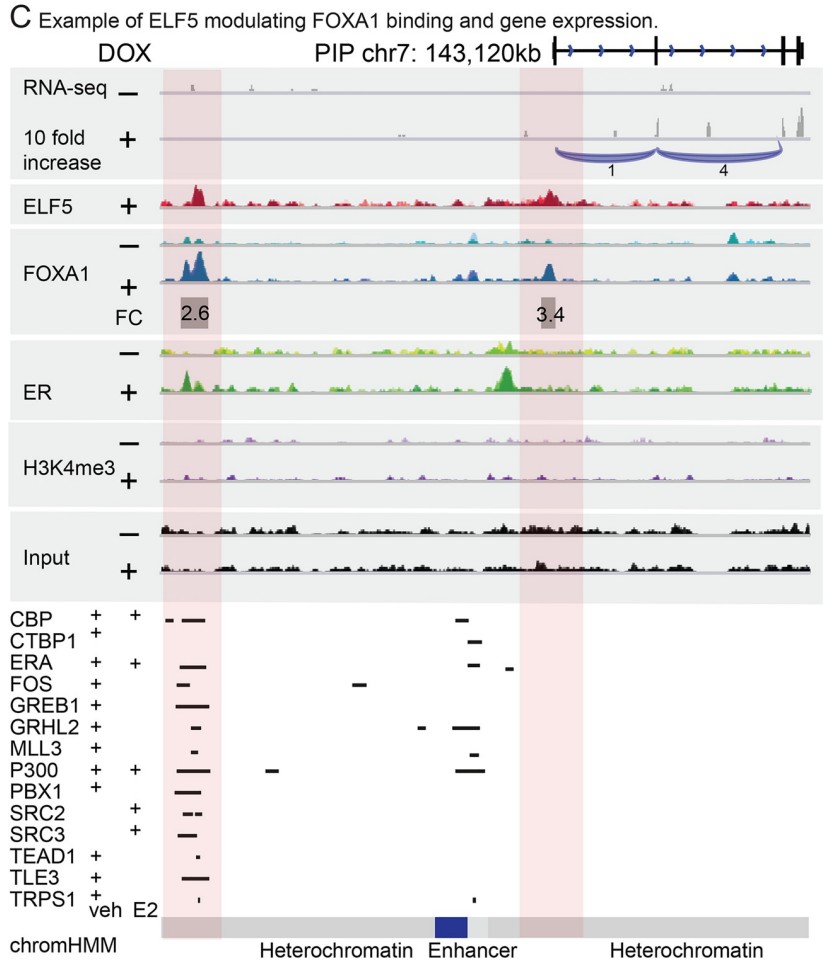

**Fig 4. Changes in ELF5, FOXA1 and ER binding to the genome are reflected in gene expression.** Panel **A**, Expression of genes identified by GREAT as ELF5, ER or FOXA1 transcriptional targets, showing their position in the distribution of differential expression (DE) in the RNA-seq data in response to induction of ELF5. GSEA-type visualization, nominal enrichment score (NES) and corresponding FDR (padj) are shown (R package fgsea). Panel **B**, as for panel A but showing genes for which FOXA1 or ER binding was increased or decreased by induction of ELF5. Panel **C**, coding-region of an ELF5-regulated gene involved in endocrine resistance (*PIP*) showing differential gene expression by RNA-seq, (exon splicing in blue) and increased FOXA1 binding by ChIP-seq at 2 locations with fold change (FC) shown, in response to induction of ELF5 with doxycycline (+) compared to vehicle (-). Binding of the indicated transcription factors to this locus from reported ChIP-seq experiments is shown together with the chromHMM model [61].

identified as one of the three top-ranking proteins by Mascot score using the very stringent criteria of present in all five replicates and absent in all five controls, along with DNA-dependent protein kinase catalytic subunit (DNA-PKcs) and protein transport protein SC16A (Fig 5A). DNA-PKcs is involved in the DNA damage response but also interacts with ER [47] and modulates ER transcriptional activity [48]. Several proteins known to interact with DNA-PKcs were identified at lower stringency, including XRCC5/Ku80 (3/5 replicates), DNA topoisomerase 2-beta (TOP2B, 2/5 replicates), with DNA topoisomerase 2-alpha (TOP2A) and poly (ADP-ribose) polymerase 1 (PARP1) present in 1/5 replicates and the second Ku sub-unit (XRCC6/Ku70) present in 2/5 replicates but also in one IgG control experiment (S1 Table). Comparison to proteins known to interact with ER [34] revealed extensive overlap and included the key ER transcription complex members GRHL2 and KDM1A (Fig 5B and S2 Table) which were also identified as interacting with ELF5 [49] in mouse trophoblasts (S3 Table).

Immunoprecipitation experiments also demonstrated the interaction between DNA-PKcs and ELF5 (S8A Fig). We used the Duolink Proximity Ligation Assay (PLA) to further test the interaction with DNA-PKcs. We observed the presence of multiple nuclear signals in response to induction of ELF5. Average signal number increased and the distribution of nuclear PLA signals per cell shifted, demonstrating an interaction between ELF5 and DNA-PKcs in MCF-7 cells (Fig 5C). This effect was also seen using a second antibody recognising a different epitope on DNA-PKcs.

We then searched for an effect of knockdown of DNA-PKcs on ELF5 driven transcription. We assembled a panel of qPCR assays for ELF5-regulated genes identified by RNA-seq that also had an ELF5 binding peak within the proximal promoter. The up-regulation by ELF5 of PIP, VTCN1 and GRHL3 in MCF7 cells was confirmed, as was the down-regulation of DKK1, MATN3, SNAI2, FILIP1L and LYN. Despite their identification in the RNA-seq experiment, ELF5 did not cause consistent changes in the expression of GDF15, STAT1, or SPDEF in this experiment. For 4 of these 8 ELF5-regulated genes, DNA-PKcs knockdown altered the ability of ELF5 to modulate their expression (Fig 5D and S9 Fig). For PIP, VTCN1 and GRHL3, loss of DNA-PKcs caused further upregulation of expression by ELF5, as well as an increase in the baseline expression (-Dox) for VTCN1 and GRHL3, suggesting an inhibitory effect of DNA-PKcs on ELF5 activity. DKK1 showed the opposite effect. We were able to find suitable antibodies to demonstrate this effect for VTCN1 by western blot (S8B and S8C Fig). Thus ELF5 interacts with DNA-PKcs and this interaction influences a portion of the transcriptional output of ELF5.

## ELF5 levels are higher in hormone resistant brain metastases compared to their matched primary tumour

To determine if these effects of ELF5 could be seen in metastases from endocrine resistant breast cancer cases we interrogated gene expression profiles of matched primary and

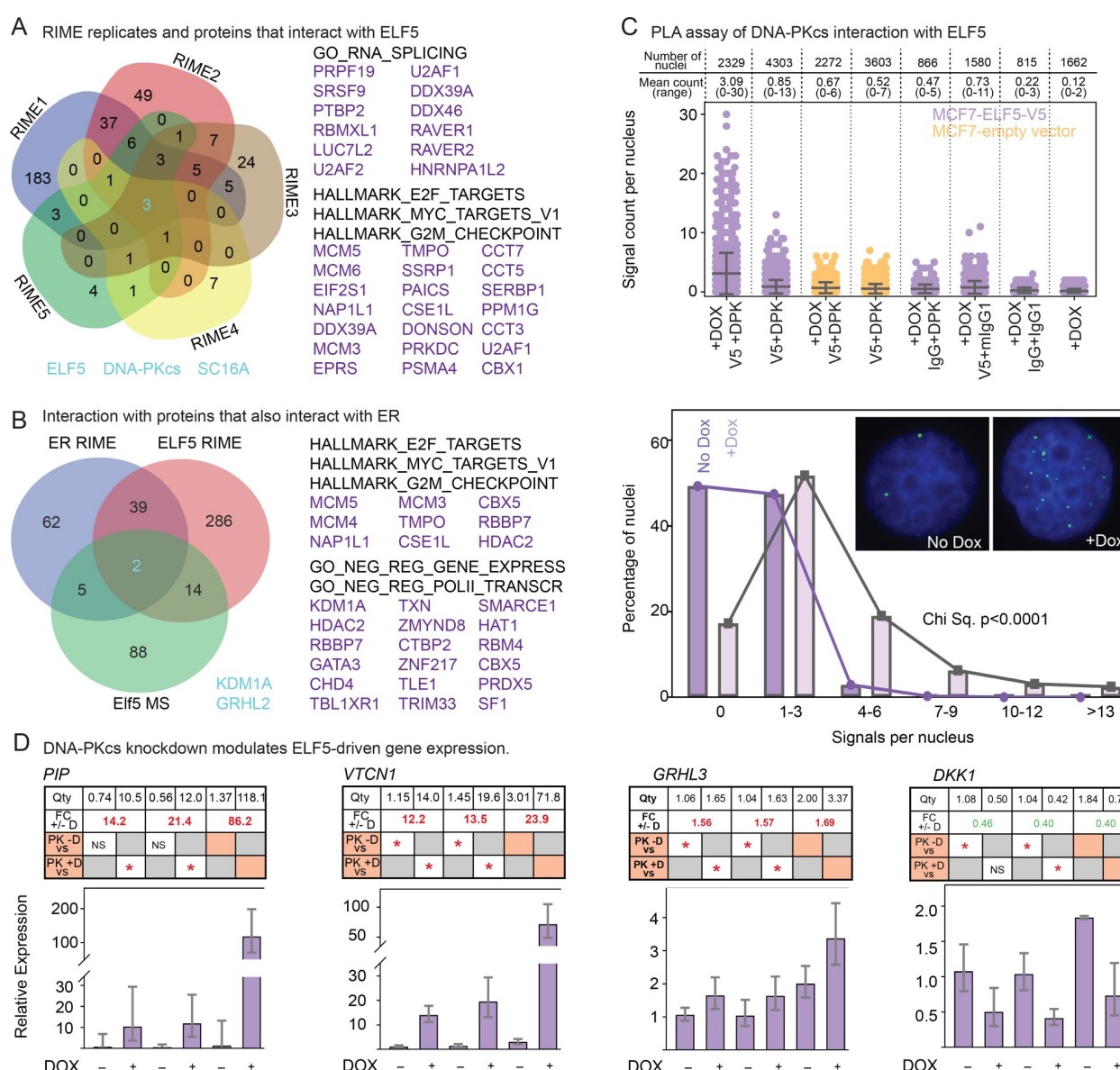

**Fig 5. Identification of ELF5 binding partners.** Panel **A**, Overlap of proteins identified in the ELF5 RIME replicates 1–5 following removal of non-specific interactors. Three proteins (ELF5, DNA-PKcs and SC16A) were identified in all replicates and no IgG controls. Gene lists show functional analysis using GSEA of interactors present in 2 of 5 replicates and not present in the IgG controls, full gene list provided in S1 Table. Panel **B**, overlap between ELF5-interacting proteins identified in at least one MCF7 ELF5 RIME replicate and those identified in mouse trophoblast cells [49], and ER RIME in MCF7 cells [34], revealing GRHL2 and KDM1A as potential common interactors. Gene lists show functional analysis using GSEA of interactors also present in ER RIME and not present in the IgG controls, full gene list provided in S2 Table. Panel **C**, top, proximity ligation assay (PLA) data showing increased signal count per nuclei between ELF5 and DNA-PKcs (DPK). Lower panel bar chart shows statistical analysis of the shift in the distribution of PLA signals per nucleus. Inset shows an example of increased PLA signal with induction of ELF5. Panel **D**, Effect of DNA-PKcs knockdown on ELF5-driven changes in expression of the indicated genes. Graphs show the mean calibrated normalised relative quantity values from three biological replicates with error bars showing 95% confidence interval. The associated table, vertically aligned with the corresponding samples in the graph, provides the exact mean normalised quantity value (Qty, row 1). Row 2 of the table indicates the effects of ELF5 induction on the target gene expression; the fold changes (FC) for the vertically aligned doxycycline induced (+D) or not induced (-D) sample pairs are shown, with red typeface indicating a significant upregulation, green a significant down regulation (one-way ANOVA). Rows 3 and 4 of the table indicate the effect of *DNA-PKcs* (PK) knockdown on the target gene expression. Row 3 compares the siPK-D sample (indicated by the orange box) with each of the untransfected (Unt) -D and the non-targeting (siNT) -D control samples, with a red asterisk indicating a significant upregulation and NS = no significant difference, one-way ANOVA). Similarly, row 4 compares the siPK+D sample (indicated by the orange box) with each of the untransfected (Unt) +D and non-targeting (siNT) +D samples.

metastatic breast cancer from brain [50]. This tumour collection contained 7 ER+ cases, 2 of which, X4 and X72, showed a large increase in ELF5 expression in the metastasis compared to the matched primary (Fig 6A). We used DESeq to search for an enriched ELF5 signature in these metastases. First, we treated all 7 of the ER+ patients as replicates, and used DESeq to rank the genes by differential expression between primary and matched metastasis. GSEA showed significant enrichment of both the set of 497 ELF5-induced genes from MCF-7 cells, and a subset of 235 genes with an ELF5 ChIP binding site within 10kb from MCF-7 (p-values of 0.00025 and 0.00038 respectively) (S7C Fig). When the patients were considered individually, three of the seven metastasis expression profiles were significantly enriched for ELF5 induced genes (X4, X62 and X72, p-values 0.00754, 0.00047 and 0.00048 respectively). When we reversed the analysis, using the MCF-7 experiments as replicates to create a ranked gene list and the differential gene expression between one primary and its matched metastasis to create the gene-sets, the results were also significant (p-values 0.00064, 0.00115 and 0.04704 respectively). This provides evidence that the effects of elevated ELF5 in MCF-7 cells also occur in three breast cancer metastases that have become resistant to endocrine therapy, consistent with elevation of ELF5 providing an additional mechanism driving acquired endocrine resistance.

In summary our experimental investigation shows that induction of ELF5 alters patterns of estrogen-driven gene expression via transcriptional mechanisms, which includes a gain of new FOXA1 and ER binding sites, regulation of the expression of members of the ER complex and direct interaction with members of the ER transcriptional complex. This finding provides a mechanistic explanation for the role of ELF5 in the modulation of estrogen action and the acquisition of resistance to endocrine therapy.

## Discussion

FOXA1 acts as a pioneer factor to regulate ER binding [8] and plays a crucial role in endocrine resistance [8, 12]. The genomic distribution of FOXA1/ER binding sites is significantly altered in endocrine resistant cell lines and poor-prognosis breast cancers [12]. Here we show that a very similar redistribution of FOXA1 binding can be caused by an increase in ELF5 levels, directly implicating ELF5 in this mechanism of endocrine resistance. Moreover, increased ELF5, and the resulting changes in FOXA1 or ER binding, regulated a set of genes with reported roles in driving endocrine resistance. For example, *CD36* increased proliferation and migration of breast cancer cells and antagonised tamoxifen effects [40]. *CUEDC1* is a transcriptional target of ER regulated via the associated CUTE enhancer discovered by CRISPR screening [51] and its product is essential for ER-driven MCF-7 cell proliferation [41]. LAMP3 is a regulator of autophagy that is induced by tamoxifen and causes insensitivity to this drug. Its expression is predictive of early disease progression. LTBP2 modulates the TGF beta pathway and is downregulated by estrogen and upregulated by TGFbeta and acquisition of the invasive phenotype. Elevated LTBP2 predicts early disease progression [44]. SDCBP is negatively correlated with ER in breast cancers and is down regulated by estrogen treatment. In ER-negative cancers it positively regulates cell cycle control [43]. *PIP* is expressed in ER+ and apocrine breast cancers and is regulated by androgens. Its product is an aspartyl protease required for RTK activation of FAK and other kinases by integrins and is involved in cancer cell invasion. It is required for proliferation in response to estrogen, but also for proliferation of tamoxifen resistant T-47D cells [45, 52].

We used RIME to search for a direct physical interaction of ELF5 with FOXA1, ER and members of the ER transcriptional complex. Limited evidence for an interaction with the histone demethylase LSD1/*KDM1A* was found. We discovered and verified interaction of ELF5

**A**  ESR1 and ELF5 expression in matched primary and brain metastases

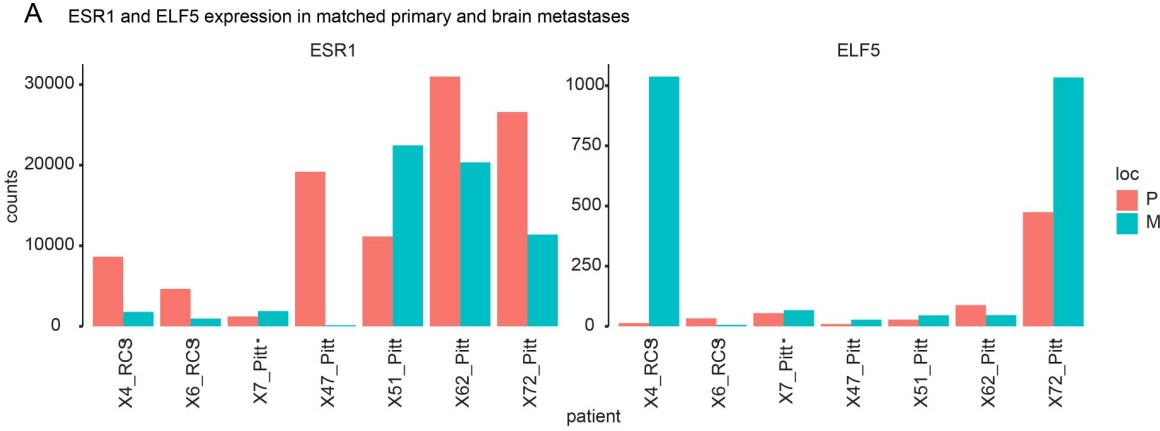

**B**  Enrichment of ELF5 expression signatures in matched metatstases compared to primaries

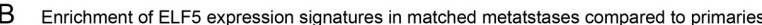

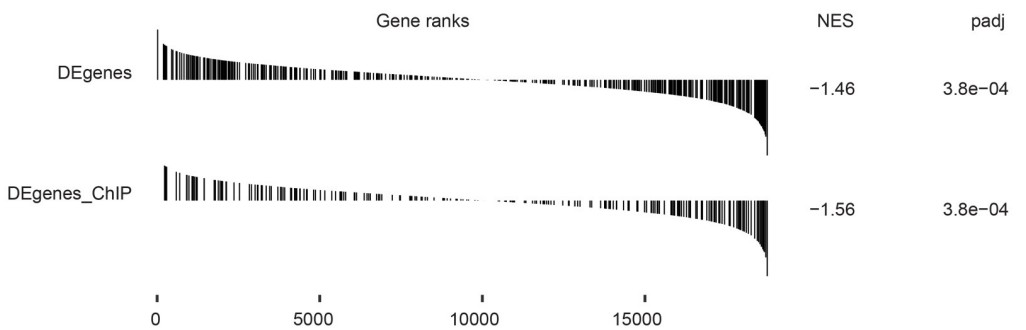

**C**  Leading edge DE genes (ELF5 CHIP targets *):

| | | | |
|---|---|---|---|
| LCN2* | FILIP1L | MXRA7 | TNFRSF19 |
| CRISP3* | APCDD1 | MME* | SDCBP* |
| IL1RN | MFSD2A* | ZFAS1* | FBLN5 |
| WISP2* | DMRTA1* | GLRX* | B4GALNT3* |
| VTCN1* | CLDN1 | SH3PXD2A | EMP1* |
| COL3A1* | B4GALT1* | GAL3ST1 | TMEM40* |
| PXDN | RPS6KA3 | FAS | KRT14 |
| COL12A1* | PLEKHG2* | CCDC160* | CTGF* |
| SNAI2* | LGALS1* | MID1* | ST3GAL1* |
| DNASE2B* | ARSJ* | RASD1 | LAMB3 |
| FUT3* | MGLL* | IVL | TMC5 |
| VGLL1 | HEG1* | ADM2 | ANTXR2 |
| COL5A1 | CHAC1 | CALD1 | |
| PLAU* | S100A9* | CHST1 | |
| GPC6* | PRICKLE1* | GRHL3* | |
| CLIC3 | MATN3 | CAMK2N1* | |
| KCNJ8 | SPNS2 | TGFBR2 | |
| FCGBP* | PAPSS2* | LAMB1 | |
| DIO2 | PDE4A* | PTGS1* | |
| LRRC15* | NUPR1 | CDC42EP5 | |
| CYBRD1 | ARHGDIB* | PTPRE* | |
| SEMA3D | RHOBTB2 | CYR61 | |
| B3GNT3* | AOX1 | KRT17 | |
| NECTIN3 | SALL4 | SLC9A2* | |
| ALDH1L2 | LOXL2 | EHHADH | |

**Fig 6. ELF5 expression is increased in endocrine resistant metastases.** Panel **A**, levels of *ESR1* and *ELF5* expression in ER+ matched primary (P) and brain metastases (M) normalised from data produced by Vareslija and colleagues [50]. Panel **B**, GSEA determined enrichment (NES) of gene sets comprising genes differentially expressed in MCF-7 cells due to induction of ELF5 (DE) or a subset with an ELF5 ChIP binding site with 10kb (DEgenes_ChIP). Panel **C**, genes at the leading edge of the enrichments shown in panel B, asterisks indicate a gene that is also a transcriptional target of ELF5 by ChIP-seq.

with ER-complex member DNA-PKcs. Whether ELF5 acts by participation in a complex with ER, or by sequestering and operating the complex with its key members in preference to ER, remains to be determined. One key unifying feature of our interaction studies is the identification of members of the CoREST complex. DNA-PKcs is a member of the CoREST complex and represses ER-driven transcription [48]. LSD-1 also participates in the CoREST complex but when bound to ER it can activate transcription associated with demethylation of H3K9me2 [53], indicating that these complexes can both repress and activate transcription.

We found an increase in ELF5 levels, and enrichment of the ELF5 transcriptional signature, in the endocrine resistant brain metastases of ER+ primary tumours. This is consistent with our proposal that in some cases of endocrine resistance, increased levels of ELF5 are driving the acquisition of insensitivity to estrogen. This is unlikely to be the sole cause of acquired endocrine resistance but may play a contributing role in many cases. We reported that RANKL induces ELF5 levels in progenitor cells, forcing their differentiation toward the alveolar lineage [17]. If this mechanism continues in breast cancer then adjuvant treatment with therapeutics such as denosumab during endocrine therapy may reduce the incidence of endocrine resistance, and a signal consistent with this action was reported by the ABCSG-18 trial (San Antonio Abs. S2-02 Can. Res. 76:2016).

In conclusion we have demonstrated that ELF5 can modulate estrogen action via modulation of the genomic regions that ER and FOXA1 occupy to cause changes in ER-driven gene expression and in particular changes in the expression of genes implicated in the acquisition of endocrine resistance. ELF5 caused alterations in the expression of ER transcriptional complex members, and altered interactions with members of the ER-coactivator complex, characterised by their participation in the CoREST complexes. Together this establishes a multi-faceted mechanism by which ELF5 can modulate and repress estrogen-driven gene expression. This provides a mechanism by which increased ELF5 expression could drive progression of ER + breast cancer to become resistant to endocrine therapy. We have previously shown that increased ELF5 can drive mammary cancer metastasis, establishing that ELF5 can cause the acquisition of the two defining features of progressive ER+ breast cancer, endocrine resistance and renewed metastasis.

## Methods

### Ethics statement

All animal experimentation was approved by the Garvan/St Vincent's animal experimentation ethics committee, approval 17–23.

### Stable cell lines and culture

MCF-7 cells expressing ELF5 Isoform 2 tagged with V5 were created and maintained as described [19]. Puromycin (Sigma-Aldrich, St Louis, Missouri, USA) at 1ug/mL was used to maintain selection. Doxycycline (Dox, Sigma-Aldrich) at 0.1ug/mL in water was used daily to induce ELF5.

### ChIP-seq

MCF7 or T-47D -ELF5-Isoform2 cells treated with Dox or vehicle from 24 hours after plating for 24 (T-47D) or 48 (T-47D and MCF-7) hours then cross-linked for 10 minutes at room temperature using 1% formaldehyde. Four independent replicates for ER, FOXA1, ELF5 and H3K4me3 ChIP-seq were performed according to the protocols described in [8] and [19].

## ChIP-seq sample preparation

MCF7-ELF5-Isoform2 cells were seeded in 15cm plates, and doxycycline (or vehicle) treatment was commenced 24 hours after plating. After 48 hours of doxycycline treatment, cells were cross-linked for 10 minutes at room temperature using 1% formaldehyde diluted in cell growth medium. After 10 minutes, formaldehyde was quenched with 0.2M glycine. Plates were then placed on ice and washed x 2 with cold PBS. Cross-linked cells were collected in 2mL PBS using a cell scraper and pellets containing approximately 20 million cells were stored at -70˚C. Four independent replicates for ER, FOXA1, ELF5 and H3K4me3 ChIP-seqs and 6 input replicates (one per lane) were performed according to the protocols described in [54], with or without Dox treatment. ELF5 was IP with a mixture of anti V5 and anti ELF5 (Santa-Cruz N20) antibodies. ELF5 samples were treated with Dox only. DNA purification following IP was performed using phenol:chloroform:isoamyl alcohol and Phase Lock Gel tubes. Libraries were prepared using the TruSeq ChIP Sample Preparation Kit (A) from Illumina, with AMPure XP bead double-sided size selection. Samples were multiplexed in 3 pooled libraries (library 1 FOXA1 samples and input, library 2 ER samples and input, library 3 ELF5, H3K4me3 samples and input) and sequenced in 3 lanes on the Illumina HiSeq 2500.

## ChIP-seq peak calling

Peaks were called using MACS 2.0.10. Consensus peaks were present in three out of four replicates (or two out of two for H3K4me3). Differential binding to peak regions was defined with R package DiffBind with default parameters [12].

## Functional annotation of peak regions

Functional analyses used the Genomic Regions Enrichment of Annotations Tool (GREAT) [25].

## Motif analysis

Enrichment of transcription factor DNA binding motifs under peaks was performed with MEME-ChIP, using default parameters [55]. Genomic regions with repetitive elements were adopted from Repbase [26].

## RNA-sequencing (MCF7-ELF5 cells)

MCF7-ELF5-Isoform2 cells treated with Dox or vehicle from 24 hours after plating for 48 hours. Sequencing used the Illumina HiSeq2000 using v3 SBS reagents and 100bp paired-end reads. Alignment was done with STAR (v 2.4.0d) [56] against the human genome (hg38) with gencode v20 annotations. Transcript counts were summarised and transcripts per million (TPM) calculated using RSEM (v 1.2.18) [57]. Counts were normalised using TMM [58] and transformed using voom [59]. Differential expression analysis was carried out using limma [60]. When performing GSEA for comparison of MCF-7 RNA-seq with T-47D and MCF-7 microarrays, FDR cut-off for DE genes from ELF5 treated T-47D and MCF-7 microarrays were 0.05.

## Patient data analysis

Raw counts of matched primary and metastatic breast cancer RNA-seq data were downloaded from https://github.com/npriedig/jnci_2018. Ranked list of genes was calculated with *DESeq* using patients as replicates, or estimating dispersion for individual patients from the rest with

method = "blind" and sharingMode = "fit-only". GSEA was performed with R package *fgsea*, with $10^5$ permutations, as described above.

## Rapid Immunoprecipitation of Endogenous Protein (RIME)

MCF7-ELF5-Isoform2 cells treated with Dox or vehicle from 24 hours after plating for 48 hours, cross-linked using 1% methanol-free formaldehyde (Thermo Fisher). RIME was performed as described [46] with modifications (see extended methods) using a 1:1 mix of anti-ELF5 and anti-V5 antibodies.

## Proximity ligation assays (PLAs)

Cells were seeded on glass coverslips in 12-well plates (Corning) and treated after 24 hours with doxycycline or vehicle for 48 hours. Three biological replicates were performed.

## siRNA transfection

ON-TARGETplus human *PRKDC* SMART pool siRNA and ON-TARGETplus non-targeting siRNA #1 (Dharmacon, Lafayette, Colorado, USA) were used. All transfections were performed using Lipofectamine RNAiMAX (Thermo Fisher). After 24 hours, the medium was changed and puromycin and Dox treatment was started on day 2 and cells were collected on day 4.

## Quantitative PCR

All qPCR reactions were run on the Applied Biosystems ABI7900 qPCR machine (Thermo Fisher). Two to three technical replicates were run for each sample, as well as negative controls (no template, no reverse transcriptase, water). Standard curves using a 1:10 dilution series were run for every assay.

Complete methods are available as supplementary material (SDoc1 Text). Raw data is available via ArrayExpress E-MTAB-7641and Proteome exchange PXD013349.

## Supporting information

**S1 Fig. Quality control measures for the ChIP-seq experiments.** MCF-7 cells stably infected with either an empty retroviral doxycycline-inducible vector, or one expressing isoform 2 of human ELF5 tagged with V5 [19] (Panel **A**), were treated with 0.1ug/ml doxycycline for 48 hours prior to cross-linking and processing for ChIP-seq using antibodies precipitating ELF5, ER, FOXA1 or H4K4me3 together with inputs. Four independent replicates for each ChIP (2 for H3K4me3) were conducted. A set of standard quality control measures were evaluated. Panel **A**, ELF5 expression level before and after induction with DOX compared to levels of induction achieved in T-47D cells with R5020 [19], or in mammary gland by 18 days of pregnancy in mice. Panel **B**, Read depth and errors, panel **C**, paired read score, panel **D** sequence duplication, panel **E** GC content, panel **F**, reads in peaks and panel **G**, signal strength. Panel **H,** principle components analysis (PCA), showed the presence of a peak in 3 of 4 replicates correctly grouped the ChIP replicates and separated all of the FOXA1 plus and minus dox replicates.
(TIF)

**S2 Fig. Additional aspects of ELF5 genomic binding.** Panels **A**, GREAT functional analysis of ELF5 genomic binding using MSigDB gene sets as indicated. Panel **B**, overlap of MCF-7 ChIP peaks with those observed in T-47D cells [19].
(TIF)

**S3 Fig. ELF5 binds to repetitive elements.** Panel **A**, sequence of motifs at ELF5 binding sites with identity to Alu repeats (DFAM) with red blue and green color bars showing the most frequent arrangements of these motifs and their enrichment (E) p value. Panel **B**, RepeatMasker analysis of repeat sequences at ELF5 binding sites showing number and type detected. Panel **C**, consensus ETS motif under ELF5 binding sites at repeats. Panel **D**, distribution of the indicated repeat types around ELF5 binding sites at the indicated window sizes. Panel **E**, odds ratios for finding the indicated repeat types under all transcription factor binding sites (wgEncode TfbsV3), under FoxA1, ER and ELF5 with or without DOX treatment, then at ELF5 binding sites within highly occupied target regions (HOT), enhancers (E), super enhancers (SE) or in the vicinity of differentially expressed genes (DE). Error bars represent standard error. (TIF)

**S4 Fig. UpSet analysis of transcription factors significantly co-located with ELF5.** UpSet analysis, using the transcription factors whose binding is most frequently co-located with ELF5, to identify patterns of co-binding at differentially expressed (DE) genes, (Panel **A** 119 genomic loci), super enhancers (Panel **B** 259 loci) and enhancers (Panel **C** 2644 loci). Numbers above the transcription factor sets show the number instances of that specific set. Black dots indicate the presence within the set of the indicated transcription factor. (TIF)

**S5 Fig. ELF5-induced gene expression analysed by GSEA and Cytoscape.** Scalable .pdf showing complete Cytoscape representation of the RNA-seq data. Each circle (node) is sized to indicate the relative number of genes in the set and coloured to show enrichment score in response to ELF5. Nodes with overlaps in their gene content are linked by green lines and are clustered according to the degree of overlap. Download and zoom to see the detail. (TIF)

**S6 Fig. ELF5-induced gene expression analysed by RNA-seq.** Panel **A**, GSEA of MSigDB Hallmark gene sets coloured according to enrichment score as indicated by the scale. Panel **B**, example GSEA plots from the MSigDB C2-all sets showing significant enrichment. Panel **C**, enriched ChIP sets (ranked by Enrichr combined score) identified in the regulatory regions of the top 100 differentially expressed MCF7-ELF5 RNA-seq genes (filtered for absolute fold-change >1.5 and ranked by FDR). The identifier for each ChIP set contains the name of the transcription factor followed by the PubMed ID, the type of experiment (ChIP-seq or ChIP-chip), the cell line or tissue, and the species. The top 10 sets (of 37 sets with an FDR <0.05) are shown. Analysis was performed using the Enrichr ChIP enrichment analysis (ChEA) tool. Panel **D**, enriched ChIP sets identified in the regulatory regions of down-regulated genes. Panel **E**, enriched ChIP sets identified in the regulatory regions of up-regulated genes. Panel **F**, enriched transcription factor motifs in ELF5 regulated genes from the TRANSFAC and JASAPR databases. No enriched motifs were identified for the down-regulated RNA-seq genes. (TIF)

**S7 Fig. Characteristics of FOXA1 binding sites enriched or depleted by induction of ELF5.** Panel **A.** Venn diagram for co-occurrence of binding peaks called present by MACS for ELF5, FOXA1 and ER. Panel **B**, matching UpSet plot for co-occurrence of binding peaks called present by MACS. Panel **C**, motif probability analysis of sequences under FOXA1 binding sites that were increased by induction of ELF5. Panel **D)**, motif probability analysis of sequences under FOXA1 binding sites that were decreased by induction of ELF5. Panel **E**, ELF5, FOXA1, ER and H3K4me3 peaks are plotted, before and after induction of ELF5, centred on all FOXA1

summits that overlap enhancers and superenhancers. Color scale denotes log 10 of MACS score. Panel **F**, consensus ETS motifs under ELF5 binding sites in the indicated genomic regions.
(TIF)

**S8 Fig. ELF5 interacts with DNA PKcs to regulate VTCN1.** Panel **A**, immunoprecipitation of ELF5 co-precipitates DNA-PKcs. Samples were prepared using the RIME protocol and immunoprecipitated with a combination of ELF5 and V5 antibodies or IgG control. Blots for V5 and DNA-PKcs are shown. Lane 1 is the input or total lysate (In), lane 2 is the ELF5-V5 immunoprecipitation (IP) and lane 3 is the IgG control immunoprecipitation (IgG). Lanes 4 and 7 are supernatants from the immunoprecipitations (Sup, representing unbound protein), while lanes 5–6 8–9 are supernatants from the first (W1) and third (W3) bead washes (indicating no residual unbound protein after the final third wash). RIME 5 is replicate 5 of the ELF5-V5 RIME experiments, while additional replicates 1 and 2 did not form part of the ELF5-V5 RIME dataset. Panel **B and C**, Western blots for MCF-7 (panel B) and T-47D (panel C) cell lines, stably modified with doxycycline-inducible pHUSH-ELF5 isoform 2 or isoform 3 vector (empty vector as a control). Cells were untransfected (Unt), transfected with a non-targeting siRNA (NT) or transfected with siRNA targeting DNA-PKcs (PK). Cells were also treated with doxycycline (Dox, indicated by + symbol) or vehicle (-). Each box represents an individual blot and is shown with the corresponding beta-actin (b-actin) loading control.
(TIF)

**S9 Fig. Knockdown of DNA-PKcs alters ELF5-driven gene expression.** Panels **A-H**, Effect of DNA-PKcs knockdown on ELF5-driven changes in expression of the indicated genes in MCF-7 and T47-D cells. Fold change is indicated in red where DNA-PKcs exerted a significant suppression of gene expression and in green where DNA-PKcs enhanced expression. Cells were untransfected (Unt), transfected with a non-targeting siRNA (NT) or transfected with siRNA targeting DNA-PKcs (PK). Cells were also treated with doxycycline (Dox, +) or vehicle (-). Graphs show the mean calibrated normalised relative quantity values from three biological replicates with 95% confidence interval. The associated table, vertically aligned with the corresponding samples in the graph, provides the exact mean normalised quantity value (Qty, row 1). Row 2 of the table indicates the effects of ELF5 induction on the target gene expression; the fold changes for the vertically aligned +Dox and -Dox sample pairs are shown, with red typeface indicating a significant upregulation (one-way ANOVA), green a significant downregulation and black a non-significant fold change. Rows 3 and 4 of the table indicate the effect of *DNA-PKcs* knockdown on the target gene expression. Row 3 compares the siPK -Dox sample (indicated by the orange box) with each of the Unt -Dox and the siNT -Dox samples, with a red asterisk indicating a significant upregulation and a green asterisk a significant downregulation (NS = no significant difference, one-way ANOVA). Similarly, row 4 compares the siPK +Dox sample (indicated by the orange box) with each of the Unt +Dox and siNT +Dox samples.
(TIF)

**S1 Table. Proteins interacting with ELF5 discovered by RIME.** Proteins are listed by the level of stringency required to include them, by progressive relaxing the requirements for presence in the replicates, as indicated.
(PDF)

**S2 Table. Proteins interacting with ELF5 compared to proteins known to interact with ER.** Comparison to proteins known to interact with ER in MCF-7 cells [34] compared to those

identified to interact with ELF5 by RIME in MCF-7 cells.
(PDF)

**S3 Table. Proteins interacting with ELF5 compared to proteins known to interact with ELF5 in trophoblasts.** Comparison of proteins found to interact with ELF5 by RIME in breast cancer cells compared to those identified as interacting with ELF5 [49] in mouse trophoblast stem cells by mass spectrometry (MS).
(PDF)

**S1 Document. Complete Methods.** Full description of methods used in PDF format.
(DOCX)

**S2 Document. MCF ChIP experiments used.** Hyperlinks to the MCF-7 ChIP data used in PDF format.
(CSV)

# Acknowledgments

The authors acknowledge the support of Ms Gillian Lehrbach, Garvan Tissue Culture Facility Sydney for advice with breast cancer cell tissue culture, Dominic Kaczorowski for DNA sequencing and Aaron Statham for DNA sequencing quality control, from the Kinghorn Center for Clinical Genomics, Sydney, and Helen Speirs from The Ramaciotti Centre for Genomics UNSW Sydney for RNA sequencing.

# Author Contributions

**Conceptualization:** Catherine L. Piggin, Alexander Swarbrick, Matthew J. Naylor, Maria Kalyuga, Warren Kaplan, Samantha R. Oakes, David Gallego-Ortega, Susan J. Clark, Jason S. Carroll, Nenad Bartonicek, Christopher J. Ormandy.

**Data curation:** Catherine L. Piggin, Daniel L. Roden, Andrew M. K. Law, Mark P. Molloy, Christoph Krisp, Samantha R. Oakes, David Gallego-Ortega, Jason S. Carroll, Nenad Bartonicek, Christopher J. Ormandy.

**Formal analysis:** Mark P. Molloy, Christoph Krisp, Nenad Bartonicek, Christopher J. Ormandy.

**Funding acquisition:** Christopher J. Ormandy.

**Investigation:** Catherine L. Piggin, Andrew M. K. Law, Maria Kalyuga, Jason S. Carroll, Nenad Bartonicek, Christopher J. Ormandy.

**Methodology:** Catherine L. Piggin, Mark P. Molloy, Christoph Krisp, Jason S. Carroll, Christopher J. Ormandy.

**Project administration:** Catherine L. Piggin, Samantha R. Oakes, Jason S. Carroll, Nenad Bartonicek, Christopher J. Ormandy.

**Resources:** Jason S. Carroll, Christopher J. Ormandy.

**Software:** Nenad Bartonicek.

**Supervision:** David Gallego-Ortega, Jason S. Carroll, Christopher J. Ormandy.

**Validation:** Nenad Bartonicek, Christopher J. Ormandy.

**Visualization:** Daniel L. Roden, Mark P. Molloy, Samantha R. Oakes, Nenad Bartonicek, Christopher J. Ormandy.

**Writing – original draft:** Christopher J. Ormandy.

**Writing – review & editing:** Catherine L. Piggin, Daniel L. Roden, Andrew M. K. Law, Mark P. Molloy, Christoph Krisp, Alexander Swarbrick, Matthew J. Naylor, Maria Kalyuga, Warren Kaplan, Samantha R. Oakes, David Gallego-Ortega, Susan J. Clark, Jason S. Carroll, Nenad Bartonicek, Christopher J. Ormandy.

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
