## [Decision Letter · Decision Letter 0]

30 Sep 2019

Dear Dr Ormandy,

Thank you very much for submitting your Research Article entitled 'ELF5 modulates the estrogen receptor cistrome in breast cancer.' to PLOS Genetics. Your manuscript was fully evaluated at the editorial level and by independent peer reviewers. The reviewers appreciated the attention to an important problem, but raised some substantial concerns about the current manuscript. Based on the reviews, we will not be able to accept this version of the manuscript, but we would be willing to review again a much-revised version. In particular, Reviewer 2 highlights several important issues that require attention. We cannot, of course, promise publication at that time.

If you decide to revise the manuscript for further consideration at PLOS Genetics, please aim to resubmit within the next 60 days, unless it will take extra time to address the concerns of the reviewers, in which case we would appreciate an expected resubmission date by email to plosgenetics@plos.org.

[LINK]

We are sorry that we cannot be more positive about your manuscript at this stage. Please do not hesitate to contact us if you have any concerns or questions.

Yours sincerely,

Matthew L. Freedman

Associate Editor

PLOS Genetics

David Kwiatkowski

Section Editor: Cancer Genetics

PLOS Genetics

Reviewer's Responses to Questions

**Comments to the Authors:**

Reviewer #1: Piggen et al., present a valuable study on the role of ELF5 in mediating endocrine resistance. Several novel aspects include: 1) ELF5 ChIP-seq, 2) Insights gained from FOXA1/ER displacement and potential role in distinct gene regulation, 3) Insight into the ELF5 protein complexes. Major comments which may strengthen the findings/conclusions of this study are:

1) It’s not clear from this study whether the protein expression levels of ELF5 are consistent with expression in tumor samples upon DOX induction. My concern is whether ELF5 expression is elevated to an extent that is not observed in disease - and therefore also resulting effects on FOXA1/ER cistrome.

2) In other disease systems, super enhancers have been used to define and predict activity of core TFs that regulate cell (disease state) identity. Are these associated with the 20+ (line 166) TFs ? or perhaps defines a distinct subset overlapping with ER-transcriptional complex members.

3) As FOXA1 is known to have pioneering activity, following ELF5 over-expression are de novo FOXA1 binding sites established at previously closed/open chromatin ?

Reviewer #2: The authors aimed to investigate the mechanism by which the alveolar lineage transcription factor ELF5 may drive endocrine therapy resistance. They use multiple ‘omics approaches to do so (with many replicates), however the data often felt like a laundry list of genes or proteins (and in some cases, only a few reproducible ones are shown) without a clear mechanism by which ELF5 acts. While the title and abstract of this paper are provoking in terms of a potential mechanism by which ELF5 can act to promote breast cancer, especially in the context of resistance to estrogen blockade, the data unfortunately fall short in actually demonstrating this. Many of the conclusions are overstated based on what the data demonstrates, the data are largely correlative, and overall lacks a clear mechanism for which ELF5 might reorganize the epigenome, and as such, the manuscript in its current state is not sufficient for publication.

1. By ChIP-seq the authors found that genomic regions of genes bound by ELF5 were also binding sites for ER and FOXA1. It is not shown anywhere what ELF5 ectopic expression looks like in either BrCa cell line, and T47D data is largely ignored throughout the manuscript. The RNA-seq data is not shown in its own right for both cell lines. These data are critical as the authors later call ELF5 DEGs. Are ER and FOXA1 regulated transcriptionally by ELF5 expression? How do we know that the ER and FOXA1 new binding sites are not merely caused by transcriptional upregulation of these TFs, which in turn causes ectopic binding? Overall, the authors need to better describe their ELF5 expression system and the RNA-seq data.

2. The authors also describe overlap of ELF5 with repetitive elements, with dedicated main figures to this point, but the context of this is completely unclear as they never come back to this finding later in the paper. What is the significance of this? Does it help to understand newly acquired binding sites? (also SINE, LINE, MIRb are not clearly defined for the reader).

3. They further studied ELF5 binding partners and suggested that it interacted with the ER transcription complex, such as DNA-PKcs. What does this interaction mean? It is unclear how it was chosen for study (besides being one of two identified through RIME) and it was not validated by co-IP. Also, the changes on gene expression upon its KD are not striking. The authors might consider other interactors to study – e.g. splicing or replications proteins (MCMs). More generally, why did the RIME data only produce 3 proteins from 5 replicates?

4. Many points are overstated in the paper based on the data presented.

-For example, the authors say at line 90 that “An ELF5 motif was present at sites of increased binding (SFig. 5C)”. Even though there is signal of ELF5 at the SFig.5C, it is not a peak or clearly biologically significant.

-The authors compare ELF5 expression in primary and metastatic patient samples. They state “This tumor collection contained 7 ER-positive cancers, 5 of which showed increased ELF5 expression”. This is an overstatement, since only 2 tumors showed increased in ELF5 expression. The others presented small increases at low levels that it might even be considered baseline. Moreover, claiming that the ELF 5 signature is significant in these patients should be validated in another patient cohort in order to make such generalizations.

-The authors also overstate the effects of ELF5 induction in some of key genes analyzed in Supp. Fig. 10, e.g. GRHL3 did not show significant regulation by ELF5.

-The authors also state at line 322 that “ELF5 diminishes the transcriptional effects of estrogen”. That again is an overstatement. To be able to draw this conclusion, the authors should perform an experiment using ELF5NoDox and ELF5Dox treated with estrogen and then compare the results to be able to conclude this.

-Also the title and main conclusions suggests reorganization of the ER cistrome, but how this occurs, which is the most interesting questions here, is not clear.

5. The Materials and Methods section falls short of how experiments were performed compromising the rigor and reproducibility of this study. For example, the authors should make clear how the dox treatment was done and what is being compared. It’s difficult to find the information whether the comparison is done on MCF-7 cells or T47D, as well as dox treatment for 24 or 48h. What do the authors actually ChIP with? ELF5 itself or via the V5 tag? What does ELF no dox look like, was it Chip’d as a negative control? Was the data normalized to this? etc, etc.

Minor points:

1. Supp Fig 8 is impossible to read leaving the reader without a clear understanding of the data presented.

2. Some of the Supp data could be brought into the main figures. The reader is constantly having to move into the Supp figures to understand the main figures. For example, some of the ChEA analysis in Supp 9A could be moved into main figures.

3. The Upset plots are sometimes very busy and difficult to follow.

4. In line 220-221 the authors compared genes up-regulated or down-regulated upon estrogen deprivation to the ELF5 gene dataset. However, they do not specify where in Fig. 3A each of these datasets are and where the data for long-term estrogen deprivation is from.

5. On line 255-256, MCMs are part of replication complexes not transcription complexes.

6. On line 258-259, the authors cite 6 genes implicated in endocrine resistance that are differentially expressed by FOXA1 or ER1, however they are not at the list presented on Fig. 3D. Are these handful of genes enough for driving the phenotype?

Reviewer #3: The authors provide a carefully done and interesting study with findings that very well support the role of the transcription factor ELF5 in playing a contributing role in endocrine resistance and breast cancer metastasis. ELF5 is known to have key roles in the development of the mammary gland and other tissues, and the authors now show that it also plays an important role in breast cancer in the acquisition of endocrine resistance and cancer progression to metastatic cancer.

To investigate the mechanism by which ELF5 engenders endocrine resistance and metastasis, the authors examine genomic binding sites (ChIP-seq) of ELF5, ER, and FOXA1, gene expression by RNA-seq in breast cancer cells with dox-inducible elevated ELF5 expression, and use RIME and PLA to identify interacting proteins and to analyze and derive information and make correlations with RNA-seq and ChIP-seq publically available datasets from breast tumors displaying endocrine sensitivity or resistance, and metastases from endocrine-resistant patients. These analyses appear to be done rigorously and well and yield valuable new information on ELF5, ER, and FOXA1 cistromes and transcriptomes in MCF-7 and T47D cells with dox-inducible increased ELF5 expression and correlations with data from clinical breast cancers.

While other studies have shown that the ER cistrome can be modulated by other nuclear receptor transcription factors including progesterone receptor and retinoic acid receptor (RAR), this is the first study to examine the impact of elevated developmental transcription factor ELF5 on the ER- and FOXA1-regulated cistrome and transcriptome in breast cancer that underlie endocrine resistance and cancer progression. Elevated ELF5 changed the cistrome binding sites of ER and FOXA1, moving them to new genomic regions similar to those seen in breast cancers that have acquired resistance to endocrine therapy. RIME and PLA showed that ELF5 interacted with components of the ER transcriptional complex including members of the CoREST complex and DNA-PKcs, and that knockdown of DNA-PKcs altered ELF5-driven gene expression implicating their relevance in ELF5 activity. RNA-seq revealed that ELF5 also changed the expression of members of the ER transcription complex and of genes known previously to drive the acquisition of endocrine therapy resistance. Overall, the studies are well done and provide mechanistic insight on the role of ELF5 in altering estrogen action and engendering endocrine therapy resistance in cancer.

Questions

1. Why is the isoform 2 of ELF5 used in the cell lines in this study?

2. Do the antibodies used detect all ELF5 isoforms, and are other isoforms present and of relevance in human tumors?

**Have all data underlying the figures and results presented in the manuscript been provided?**

Reviewer #1: Yes

Reviewer #2: Yes

Reviewer #3: Yes

PLOS authors have the option to publish the peer review history of their article (what does this mean?). If published, this will include your full peer review and any attached files.

Reviewer #1: No

Reviewer #2: No

Reviewer #3: No

---

## [Decision Letter · Decision Letter 1]

20 Nov 2019

Dear Dr Ormandy,

We are pleased to inform you that your manuscript entitled "ELF5 modulates the estrogen receptor cistrome in breast cancer." has been editorially accepted for publication in PLOS Genetics. Congratulations!

Yours sincerely,

Matthew L. Freedman

Associate Editor

PLOS Genetics

David Kwiatkowski

Section Editor: Cancer Genetics

PLOS Genetics

Comments from the reviewers (if applicable):

Reviewer's Responses to Questions

**Comments to the Authors:**

Reviewer #3: The revised manuscript is strong and has answered reviewer questions very well and provided new information requested.

**Have all data underlying the figures and results presented in the manuscript been provided?**

Reviewer #3: Yes

PLOS authors have the option to publish the peer review history of their article (what does this mean?). If published, this will include your full peer review and any attached files.

Reviewer #3: Yes: Benita S. Katzenellenbogen

**Data Deposition**

http://datadryad.org/submit?journalID=pgenetics&manu=PGENETICS-D-19-01119R1

**Press Queries**

---

## [Editor Report · Acceptance letter]

18 Dec 2019

PGENETICS-D-19-01119R1 

ELF5 modulates the estrogen receptor cistrome in breast cancer. 

Dear Dr Ormandy, 

We are pleased to inform you that your manuscript entitled "ELF5 modulates the estrogen receptor cistrome in breast cancer." has been formally accepted for publication in PLOS Genetics! Your manuscript is now with our production department and you will be notified of the publication date in due course.

With kind regards,

Nicholas White

PLOS Genetics

On behalf of:
